# Yttrium-Iron Garnet Magnetometer in MEG: Advance towards Multi-Channel Arrays

**DOI:** 10.3390/s23094256

**Published:** 2023-04-25

**Authors:** Ekaterina Skidchenko, Anna Butorina, Maxim Ostras, Petr Vetoshko, Alexey Kuzmichev, Nikolay Yavich, Mikhail Malovichko, Nikolay Koshev

**Affiliations:** 1CNBR, Skolkovo Institute of Science and Technology, 121205 Moscow, Russia; a.butorina@skoltech.ru (A.B.); n.yavich@skoltech.ru (N.Y.); n.koshev@skoltech.ru (N.K.); 2M-Granat, Russian Quantum Center, 121205 Moscow, Russia; max@rqc.ru (M.O.); p.vetoshko@rqc.ru (P.V.); a.kuzmichev@rqc.ru (A.K.); 3Laboratory of Magnetic Phenomena in Microelectronics, Kotelnikov Institute of Radioengineering and Electronics of RAS, 125009 Moscow, Russia; 4Computational Geophysics Lab, Moscow Institute of Physics and Technology, 141701 Dolgoprudny, Russia; malovichko.ms@mipt.ru

**Keywords:** gradiometers, magnetometers, MEG, OPMs, simulation, SNR, SQUID, YIGM

## Abstract

Recently, a new kind of sensor applicable in magnetoencephalography (MEG) has been presented: a solid-state yttrium-iron garnet magnetometer (YIGM). The feasibility of yttrium-iron garnet magnetometers (YIGMs) was demonstrated in an alpha-rhythm registration experiment. In this paper, we propose the analysis of lead-field matrices for different possible multi-channel on-scalp sensor layouts using YIGMs with respect to information theory. Real noise levels of the new sensor were used to compute signal-to-noise ratio (SNR) and total information capacity (TiC), and compared with corresponding metrics that can be obtained with well-established MEG systems based on superconducting quantum interference devices (SQUIDs) and optically pumped magnetometers (OPMs). The results showed that due to YIGMs’ proximity to the subject’s scalp, they outperform SQUIDs and OPMs at their respective noise levels in terms of SNR and TiC. However, the current noise levels of YIGM sensors are unfortunately insufficient for constructing a multichannel YIG-MEG system. This simulation study provides insight into the direction for further development of YIGM sensors to create a multi-channel MEG system, namely, by decreasing the noise levels of sensors.

## 1. Introduction

### 1.1. History

Magnetoencephalography (MEG) is one of the modalities for neuroimaging that allows the visualization and study of the electrical activity of the brain. The MEG technique is based on measuring the magnetic induction produced by cortical currents, using an array of sensors (magnetometers) located outside of the head. Due to the fact that biological tissues are transparent to magnetic fields, MEG provides much higher spatial resolution and frequency sensitivity than the electroencephalography (EEG) technique. On the other hand, magnetic fields are a direct consequence of electric currents, allowing for MEG to outperform methods based on magnetic resonance imaging (MRI) in terms of temporal resolution. To summarize, the MEG technique combines high temporal and high spatial resolutions, as well as good sensitivity to high-frequency brain currents, making it the most powerful among non-invasive neuroimaging tools.

The main technological difficulty of MEG is caused by the weakness of magnetic fields inside the region of measurement. The primary cortical currents are of the nA-order. Taking into account the fast decay of the magnetic field with the distance from a source, in the measurement region, it has the strength of the fT-order. This requires the use of extremely sensitive magnetic sensors and magnetic screens, such as a magnetically shielded room (MSR), with a high screening coefficient.

MEG was first demonstrated by D. Cohen et al. [1] in 1971, who proposed the use of superconducting quantum interference devices (SQUIDs) to measure brain-induced magnetic fields. Since then, for almost half a century, SQUIDs were the only sensors capable of registering MEG signals. In different SQUID systems, the scalp-to-sensor distance varies within 2–3 cm, but it is no less than 2 cm. The sensitivity of the SQUID systems reaches 3–5 $fT/\sqrt{Hz}$ [2]. However, SQUID sensors have several significant drawbacks related to the demand for cooling them with liquid helium to maintain the superconductivity phenomenon. That is why such SQUID sensors are called low-Tc SQUIDs. SQUID-based devices are very expensive and bulky. Moreover, SQUID helmets are not adaptable to different head shapes and sizes, which leads to the appearance of additional errors in measurements. The bulkiness of the SQUID-based MEG systems requires the use of big and heavy MSRs that are expensive themselves. All of these disadvantages explain the relatively low popularity of MEG despite its high capacity.

Recent technological advances in the field of high-temperature superconductivity (high-Tc SQUID) as well as in the field of atomic optically pumped magnetometers (OPMs) may partially lift some limitations of the conventional SQUID-based MEG technique, sufficiently spreading the popularity of MEG.

MEG systems based on high-Tc SQUID sensors are also bulky; however, the technology is quite promising. They operate at a liquid nitrogen temperature instead of helium, and while they have higher noise levels than low-Tc SQUIDs, in some exceptional cases, it has been possible to produce high-Tc SQUIDs with a sensitivity of around 10 $fT/\sqrt{Hz}$ [3]. This fact makes the sensors much cheaper in terms of both CAPEX and OPEX (capital expenditure and operating expenditure). The high-Tc SQUIDs can also be placed much closer to the scalp (about 1–2 mm), which has advantages, as demonstrated in [4,5].

In the last decade, the MEG market has seen the emergence of OPMs. These sensors were demonstrated to be feasible for MEG purposes in 2010 [6], and they are currently the main candidate for the future of MEG, as evidenced by the significant amount of OPM-driven research in the field. Unlike SQUIDs, OPMs do not require cooling to cryogenic temperatures for operation and can be assembled into flexible arrays around the head, adapting to any head size and shape and providing a small scalp-to-sensor distance (about 6–6.5 mm). These advantages have led to the development of numerous laboratory and commercial OP-MEG systems with a sensor sensitivity level of less than 15 $fT/\sqrt{Hz}$ ([7,8,9,10,11], etc.). Indeed, the appearance of OPMs breathed new life into MEG, and OP-MEG studies are being continued [12,13,14,15]. Along with the emergence of new types of sensors, a number of simulation studies have appeared to explore and find new layouts [16,17,18,19,20]. Nevertheless, OPMs have some disadvantages, such as a narrow dynamic range (±1.5–5 nT [21,22,23]), heating problems, and a limited lifetime of the sensors. However, there is some preliminary research on perspective 4He OPMs, which could reach a sensitivity better than 50 $fT/\sqrt{Hz}$ [24,25,26]).

However, a significant step forward in MEG evolution was recently achieved in 2021 when a new type of highly sensitive sensor (with a sensitivity level of 35 $fT/\sqrt{Hz}$) was presented and demonstrated to be feasible for MEG applications [27]. The sensor is based on yttrium-iron garnet thin films and operates on the principle of a flux-gate magnetometer. The yttrium-iron garnet magnetometer (YIGM) is a solid-state sensor, and unlike SQUIDs and most OPMs, it operates at room temperature, is durable, and has a comparatively wide dynamic range (10 μT [28]). These advantages suggest the emergence of a promising new YIG-MEG technique in the near future.

### 1.2. Motivation

Regardless of the nature of novel sensor types, the development of MEG systems based on them demands a considerable amount of research in the field of system effectiveness and information capacity. This task is non-trivial and has been previously investigated by researchers worldwide. It requires the application of mathematical modeling and information theory.

The amount of information gathered by an array of magnetometers depends on several factors, such as the shape and dimensions of the sensors, the distance between the sensors and the scalp, the orientation of sensitive axes in the magnetometers relative to the orientation of sources, the particular features of the cognitive experiments, the sensitivity and intrinsic noise of the sensors in the array, and the positions of the sensors in the array [29]. Some magnetometers are designed to measure only the radial (normal) component of the magnetic induction, while others can provide measurements of two or even three orthogonal components [29], although commonly with lower sensitivity [23].

The current work explores the potential of multi-channel YIGM sensor arrays in MEG. This research is motivated by the unique nature of the sensor and its specific geometry. The zero-generation of the YIGM is quite large in size (38 × 38 × 3 mm^3^) and is noisy—the level of intrinsic noise is about 35 $fT/\sqrt{Hz}$ [27]. In [28], the authors suggest that it may be possible to decrease intrinsic noise to 1 $fT/\sqrt{Hz}$ (the theoretical limit) and reduce the size by increasing the pumping frequency. However, decreases in the size and intrinsic noise demand a large amount of engineering work of a different kind. Thus, we need to define the optimal development strategy for new generations of YIGM sensors.

Due to its flat design, the YIGM can be placed in different orientations to measure either the normal (radial) or tangential components of the magnetic induction with respect to the head (see Section 4.1). Since measuring the tangential component requires a closer distance between the sensor and scalp compared to measuring the normal component (see [27]), it is reasonable to investigate and compare the quality and amount of information that can be obtained in both ways. The information quality and capacity have been previously discussed in other research studies.

Iivanainen et al. [2] demonstrated that volumetric Ohmic currents have a greater impact on the tangential component of magnetic induction compared to the radial component. However, the claim that radial components are more robust to skull modeling errors, better at representing dipoles, and require less computational complexity than tangential components, has no theoretical basis according to [30]. Furthermore, a simulation study using realistic conductivity skull models [31] shows that non-idealities in the conductor’s shape cause equal distortions in both the radial and tangential components. The work in [32] supports this idea and shows the insignificance of extra computational complexity, i.e., of using purely tangential components with respect to an inverse problem solution.

The dependence of information capacity on the scalp-to-sensor distance has been explored by different research teams due to recent technological advances in the field of high-Tc SQUIDs and OPMs. For example, two experimental research studies [3,4] compared the responses registered by conventional low-Tc SQUID-MEG systems with those of high-Tc magnetometers placed several millimeters from the scalp, showing a considerable increase in the signal-to-noise ratio for the latter. The analysis of the lead-field matrix [2,33] and inverse problem solution [34,35] has also shown the significance of placing sensors closer to the head in terms of estimating the signal-to-noise ratio, total information capacity, and spatial resolution of the inverse problem solution.

In this paper, we will explore lead-field matrices for different possible YIGM multi-channel layouts, analyze the metrics according to the information theory, and compare them with the metrics for corresponding layouts for arrays of SQUID and OPM sensors.

## 2. Theory

This section is dedicated to brief physical and mathematical descriptions of the phenomena and techniques underlying the MEG method.

### 2.1. Governing Equations

In order to model electric and magnetic fields produced by cortical currents, the head is considered to be a closed, piece-wise homogeneous volume conductor ([36,37,38], etc.) $\Omega \in R^{3}$ with boundary ∂Ω. The interfaces inside this volume are indexed with index s=0,…,S and denoted by $\partial \Omega_{s}$. The electric current density can be presented in two parts:(1)$$J\left(x\right)=J_{p}\left(x\right)+J_{e}\left(x\right)\equiv J_{p}\left(x\right)+\sigma \left(x\right)E\left(x\right),x\in \Omega ,$$
where $J_{p}\left(x\right)$ represents the primary currents occurring in dendrites and $J_{e}\left(x\right)=\sigma \left(x\right)E\left(x\right)$ is the Ohmic volume currents driven by the non-zero electric field *E* in a volume conductor.

Due to comparably low frequencies of the sources, the quasi-stationary approximation can be applied, which allows reducing the Maxwell equations to two Poisson-like elliptic governing equations for electric and magnetic fields [39,40]:(2)$$\nabla \cdot \left(\sigma \left(x\right)\nabla U\left(x\right)\right)=\nabla \cdot J_{p}\left(x\right);$$
(3)$$\Delta B\left(x\right)=-\mu_{0}\nabla \times \left(J_{p}\left(x\right)-\sigma \left(x\right)\nabla U\left(x\right)\right),$$
where U(x),x∈Ω is the electric potential, σ(x) represents the tissue conductivity at the point x∈Ω, and $\mu_{0}$ is the vacuum permeability.

Finally, after integrating the equation for *B* and transforming it using integral identities [38], we end up with the following formula for magnetic induction:(4)$$B\left(x\right)=\frac{\mu_{0}}{4\pi }\int_{\Omega }J_{p}\left(\xi \right)\times R\left(x,\xi \right)d\xi +\frac{\mu_{0}}{4\pi }\sum_{s}\Delta_{s}\int_{\partial \Omega_{s}}U\left(\xi \right)n_{s}\left(\xi \right)\times R\left(x,\xi \right)d\xi ,$$
where $n_{s}\left(\xi \right)$ is the normal to the surface interface $\partial \Omega_{s}$ at the point $\xi \in \partial \Omega_{s}$, $\Delta_{s}$ is the difference in tissue conductivity between volumes divided by the interface $\partial \Omega_{s}$, and R(x,ξ) is the vector function introduced for brevity:(5)$$R\left(x,\xi \right)=\frac{x-\xi }{{\left|x-\xi \right|}^{3}}.$$

Thus, the magnetic induction can be calculated via simple integration after computing the electric potential using Equation (Equation 2) with proper boundary conditions. The output of the sensor is typically obtained by integrating the magnetic induction over the sensitive volume of the sensor under consideration (see Section 4.2).

### 2.2. Lead-Field Matrix

The natural discretization of Equation (Equation 4) allows obtaining the sensor outputs for $N_{s}$ sensors in the simple linear approximation [37,41]:(6)b=Lj,
where vector $b\in R^{N_{s}}$ represents the scalar outputs of $N_{s}$ sensors, $j\in R^{N_{d}}$ is the vector representing current dipoles at $N_{d}$ discrete points on the cortex, and $L\in R^{N_{s}\times N_{d}}$ is the lead-field matrix obtained with MRI-based head geometry, proper conductivities, and taking into account the integration over the sensors’ working body and the geometry of its sensitive axes. The rows of the lead-field matrix represent the outputs of sensors for unit sources stored in columns; $L_{ji}$ is the output of the *j*th sensor if only the *i*th unit source is active.

The construction of the lead-field matrix *L* is the most common way to solve the MEG forward problem. The overall number of dipoles $N_{d}$ depends on the desirable accuracy of source localization.

Based on *L* measurements, one may compute the estimation of the underlying sources, *j*, i.e., solve the *MEG inverse problem*. The inverse problem is ill-posed according to Hadamard due to its solution being unstable, and underdetermined due to $N_{s}<N_{d}$. There are different ways to solve this problem, the main ones are beamformer [42], MNE [37], LORETA [43], and MCE [44]. In the current research, we used the most conventional way to do it: minimum norm estimation, or MNE. This method is close to Tikhonov’s regularization and is implemented in almost every library or package for the solution of the MEG inverse problem. In the paradigm of MNE, the source estimate can be presented as follows [41]:(7)$$\tilde{j}=L^{T}{\left(LL^{T}+\lambda^{2}C\right)}^{-1}b,$$
where λ>0 is the regularization parameter limiting the $l_{2}$-norm of the solution, and *C* is the noise covariance matrix. Due to [2,45], the regularization parameter may be computed using the estimated signal-to-noise ratio (SNR) via the following expression:(8)$$\lambda^{2}=\frac{tr(\tilde{L}{\tilde{L}}^{T})}{N_{s}SNR}.$$

Here, $\tilde{L}=C^{-1/2}L$ denotes the lead-field matrix after the whitening transformation.

## 3. Methods

The current research was inspired by [2,33]. The first work proposes an adaptation of Shannon’s channel capacity to MEG sensor arrays, forming the metric called “information capacity”. The second work considers multiple metrics, including topography power, SNR of source, relative signal power, relative SNR, analysis of the point spread functions, and total information.

For the first research study related to YIGM array information quality, we computed the relative signal power, relative SNR, and total information capacity. For convenience, we briefly describe these metrics below in the current section.

All of the metrics presented below take into account the sensor’s noise covariance matrix *C*. In this work, we exclude all noises except the intrinsic sensor noise. Previous research in the field of yttrium-iron garnet magnetometers [28,46,47] does not demonstrate any cross-talk phenomenon, while the cross-talk of the most sensitive sensor presented in [27] will be investigated during further studies. In the current paper, since we do not have enough information about any cross-talk, we exclude it from our computations, which makes the sensors independent of each other. In this case, the noise covariance matrix $C_{YIGM}$ takes the form of a diagonal matrix with the dispersion of intrinsic noise $\sigma_{YIGM}^{2}$ on the diagonal. We note that the same reasoning is applicable to other sensor types under consideration in this work. Thus, we assume the matrices $C_{SQUID}$ and $C_{OPM}$ to be diagonal matrices with dispersions, $\sigma_{SQUID}^{2}$ and $\sigma_{OPM}^{2}$, respectively. The σ values are presented in Table 1 in Section 4.

As an anatomical model, a single high-resolution structural T1-weighted MRI scan was acquired by using a 1.5 T Toshiba ExcelArt Vantage scanner with the repetition time (TR) = 12 ms, echo time (TE) = 5 ms, flip angle = 20°, 160 sagittal slices, slice thickness = 1.0 mm, and voxel size = 1.0 mm × 1.0 mm × 1.0 mm. The cortical matter was segmented, and the estimated border between gray and white matter was tessellated. The single-layer boundary-element model was created by using watershed segmentation algorithms (FreeSurfer 4.3 software; Martinos Center for Biomedical Imaging, Charlestown, MA, USA) to reconstruct the brain’s cortical gray matter surface. For each hemisphere, the model contained 4098 dipole elements (sources), and free source orientations were used.

### 3.1. Signal Power and Signal-to-Noise Ratio

Since the *i*th column of the matrix *L* defines the output of all $N_{s}$ sensors with respect to the *i*th unit source location, it represents the magnetic field pattern of the *i*th source, and is called the source topography metric ($t_{i}$). The source topography allows estimating the sensitivity in terms of SNR, using the representation (Equation 8). Indeed, by introducing the value $q^{2}$ of source variance for independent sources, the SNR of the *i*th source can be explicitly obtained in the form [2]:(9)$$SNR_{i}=\frac{q^{2}tr\left(\tilde{t_{i}}{\tilde{t_{i}}}^{T}\right)}{N_{s}},$$
where $\tilde{t_{i}}$ means the whitened source topography: $\tilde{t_{i}}={\left(C^{-1/2}L\right)}_{i}$. Thus, the SNR metric has the same size as the source space and, therefore, allows obtaining a convenient brain map in terms of the sensitivity of the sensor array with respect to sources located in different areas of the cortex. The $l_{2}$-norm of the source topography squared, $\left|\right|t_{i}{\left|\right|}^{2}$, is defined as the topography power or signal power. Equation (Equation 9) can be simplified when the sensor noise variance $\sigma^{2}$ is equal for all sensors in the array. In this case, the topography power is linearly proportional to the SNR of source [2]:(10)$$SNR_{i}=\frac{q^{2}\left|\right|t_{i}{\left|\right|}^{2}}{N_{s}\sigma^{2}}.$$

Speaking of the relative signal power and relative SNR for two different sensor arrays, they can be calculated in the following way [2]:(11)$$\frac{SNR_{i}^{a}}{SNR_{i}^{b}}=\frac{N_{s}^{b}\sigma_{b}^{2}\left|\right|t_{i}^{a}{\left|\right|}^{2}}{N_{s}^{a}\sigma_{a}^{2}\left|\right|t_{i}^{b}{\left|\right|}^{2}}=\frac{N_{s}^{b}\sigma_{b}^{2}}{N_{s}^{a}\sigma_{a}^{2}}S_{i}^{a,b},$$
where $\left|\right|t_{i}^{a}{\left|\right|}^{2}$ and $\left|\right|t_{i}^{b}{\left|\right|}^{2}$ are the topography power of the same source *i* for the arrays *a* and *b*, respectively, and $S_{i}^{a,b}$ is the relative signal power.

### 3.2. Information Capacity

The total information capacity related to the original Shannon channel capacity was first adapted for MEG measurements by Kemppainen and Ilmoniemi [48] and then by Schneiderman [33]. In the current research, the information capacity was computed using the approach proposed in [2], using the orthogonalized SNR of the channels $SNR_{i}^{\prime }=q^{2}\left|\right|{\left(U^{T}\tilde{L}\right)}_{i}{\left|\right|}^{2}$. Here, the vectors *U* are the eigenvectors of the matrix $\tilde{L}{\tilde{L}}^{T}$. The information capacity can be calculated as follows:(12)$$I=\frac{1}{2}\sum_{i=1}^{N_{s}}log_{2}\left(SNR_{i}^{\prime }+1\right).$$

As opposed to the source power and SNR, this metric is integral and scalar, and enables the estimation of the effectiveness of the sensor array in terms of the overall information quality gathered by it.

## 4. Sensors

### 4.1. Description of the Sensors

In the current paper, we use three types of sensors for our computations: YIGM, OPM, and SQUID. The YIGM sensor is a solid-state magnetometer based on specially shaped yttrium-iron garnet thin films. As mentioned earlier, the YIGM sensor is flat by design and has dimensions of 38 × 38 × 3 mm^3^. The sensor has two perpendicular sensitive axes, both lying in the sensor’s plane. There are two possible sensor placements with respect to the head: normal to the head surface (Figure 1a), where the ’normal to the head’ coincides with one of the sensitive axes, and tangential (Figure 1b).

In the first case, it is possible to register radial (normal) and one of the tangential components of magnetic induction (in the sensor’s plane). In the second case, we can register one or two tangential components equidistant from the head. We stress that, here, the distance between the nearest housing of the sensor and the surface of the head (scalp) can be minimized down to units of millimeters. Since the magnetic field is rapidly decaying, we assume we will obtain better SNR in YIGM tangential measurements rather than in normal ones.

Taking into account the absence of the evident cross-talk phenomenon in previous YIGM measurements, it could be possible to construct the gradiometer using two YIGM sensors. Here, the sensors are placed parallel to each other with 5 mm sensor-to-sensor distances in the “normal” configuration with respect to the head (see Figure 2). In this case, the spatial tangential derivative of the normal component and one of the tangential components of the magnetic induction vector can be obtained. It is, however, impossible to prove that this scheme is feasible for measuring normal derivatives of the tangential components. Thus, in this research, we only use the scheme of the gradiometer measurements presented in Figure 2. For comparison, the same figure contains the scheme for gradiometers implemented in the reference SQUID system.

In our study, we also use QZFM OPMs (QuSpin Inc., Louisville, CO, USA) and the SQUID-system Elekta Neuromag (Elekta Oy, Stockholm, Sweden). Their parameters are reflected in numerous literature studies (see, e.g., [2,8,10,49]). The parameters for all three sensor types are compiled in Table 1. Here, by size, we mean the size of the sensor-sensitive element or working body. In the case of OPMs, it is a cubic glass cell inside the sensor itself, while in the cases of SQUID and YIGM, it is a plane. The intervals presented in this column represent the different SQUID machines and YIGM sensors of different sizes used in this paper (see details in Section 4.2). Correspondingly, the distance to the scalp *d* represents the distance from the center of the glass cell in the case of OPMs, the distance from the coil plane for SQUID, and the distance from the nearest housing in the case of YIGM. The 10 mm distance for OPMs is due to the reported heat problems with the sensors, as discussed in [50,51], and to ensure the comfort of the subjects.

### 4.2. Output Computation

For each sensor type, the sensor output was obtained by the approximate integration over the sensor’s working body, using the set of certain integration points within it. Since both SQUID and OPM sensors are considered to be well-established for MEG purposes, for these sensor types, the integration points (as well as some other parameters) are fixed in different packages related to MEG/EEG. In the current research, the forward problem solution (the lead-field matrix) was computed in MNE-Python [52], using the points listed in the file “coildef.dat”. In short, we do not list the coordinates of these points here, stressing only the number of them (shown in the fourth column of Table 1).

Taking into account the linear dimensions of the zero-generation of the YIGM sensor, the magnetic field could differ a lot over the sensor’s working body (plane), especially in normal component measurements (see Figure 1a). To decrease the integration error, we apply a rather dense mesh for integration: 5×5 points, $N_{i}=25$. The nodes of this mesh are distributed uniformly over the integration plane. The coordinates of the integration points, therefore, depend on the linear size of the sensor. In this paper, we test four different possible sizes. The current one is 38×38 mm^2^, while shortly, we are planning to decrease it to 30 × 30 mm^2^ or 20 × 20 mm^2^. We will also consider the theoretically possible size of 10 × 10 mm in the future. Since the YIGM sensor is novel, the information about its integration points was added to the “coildef” file in the standard manner. Again, for brevity, we do not list the coordinates of these integration points here.

The sensor coordinates and orientations were stored in the MNE.info structure for each sensor type and each layout listed below in Section 4.3.

### 4.3. Sensor Arrays

To compare the different layouts of the sensors, we chose two schemes of sensor locations. The first one was taken from the existing and well-established MEG system Elekta Neuromag (we refer to it as a ‘SQUID-like layout’). The second one is defined using the uniform distribution of sensors over the scalp (the area of measurement).

#### 4.3.1. SQUID-like Layout

To obtain the layout implemented in the Elekta system, we first applied typical digitization. We then defined the point on the scalp such that the line connecting it with the center of the SQUID sensor was aligned with the normal to the head surface (see Figure 3). After that, OPM and YIGM sensors were placed on this line at a specific distance for each sensor stated in Table 1. This procedure was repeated for the rest of the points obtained during the digitization stage.

In this manner, we obtained three SQUID and OPM layouts used for further computations together with a set of YIGM layouts. The original SQUID system provides us with two SQUID layouts, namely, containing 102 magnetometers (see Figure 4a) and 204 planar gradiometers (not shown). For OPMs, we obtained the SQUID-like layout consisting of 102 points, which are locations for centers of OPM gas cells (Figure 4b). For YIGM, we formed three layouts, one consisting of 102 normally oriented YIGMs (Figure 4c), one for tangential component measurements (102 tangentially oriented YIGMs), see Figure 4d), and the last one for gradiometer measurements of the radial component (see Figure 4e).

We use a 2 mm distance from the scalp to the nearest edge or plane of the YIGM sensor. Speaking of the distance between the scalp and the center of the sensor plane, it is the same 2 mm for tangential measurements (we use the sensor size of 20 × 20 mm to avoid intersections). In the case of normally oriented sensors, it depends on the size of the YIGM sensor as d=2.0+size/2 (mm). Since we worked with squares with sides of 10,20,30, and 38 mm, these distances are 7,12,17, and 21 mm, respectively. For brevity, in Figure 4 we show only YIGM layouts generated for the YIGM plane that is 20×20 mm in size.

#### 4.3.2. Uniform Layouts

Another type of sensor layout used in this study was originally designed for our implementation of the OP-MEG system [27]. In this layout, sensors were evenly distributed over the scalp area of measurement. The coordinates were selected to ensure that YIGM sensors, placed for tangential measurements, did not intersect with each other. Thus, the number of sensors in the layout depends on the sensor size. Therefore, we explore not just one layout, but a set of layouts, with the main parameters shown in Table 2. As with the SQUID-like layout, we used a 2 mm distance from the scalp to the sensor’s edge for YIGMs. The sensors, along with their integration points for these layouts, are presented in Figure 5.

## 5. Results

In order to apply the metrics mentioned in Section 3, we compute the lead-field matrices for all layouts listed in Section 4.3.

The source power for each layout was computed using the column-wise squared norm of the lead-field matrix. The SNRs were then calculated using Formula (Equation 9). For the conventional Elekta Neuromag MEG system, the source power and SNR were computed separately for magnetometers and gradiometers. In the case of YIGM, we calculated the SNRs for noise varying from 1 $fT/\sqrt{Hz}$, which is the theoretical limit, to 35 $fT/\sqrt{Hz}$, which is the current noise level of YIGMs obtained in [27].

Each layout of both OPM and YIGM was compared to the reference Elekta Neuromag layout using representation (Equation 11), forming relative signal powers and relative SNRs. For YIGMs, which had different noise levels, we formed mean SNRs. In order to compare layouts fairly, we used the same $q^{2}$ value in each calculation, chosen in such a way that the mean SNR of Elekta is equal to 1.

We also computed the channel information capacity for all layouts using Formula (Equation 12). Again, in the case of YIGM, it depends on the noise level for each layout.

### 5.1. SQUID-like Layouts

Firstly, we explore the SQUID-based layouts described in Section 4.3. The dependencies of relative SNR and the channel information capacity are shown in Figure 6. These computations were conducted using the noise levels presented in Table 1. The YIGM size in the simulation shown in Figure 6 is 20 mm × 20 mm. In order to make the plots more informative, we used the logarithmic scale along the vertical axis.

The left column of the figure presents the metrics for SQUID-based layouts consisting of magnetometers. The solid magenta line shows the performance of the original Elekta Neuromag layout, while the dashed green line represents the performance of the SQUID-based layout formed by OPMs. The solid blue, orange, and green lines represent the YIGM layouts. Please note that the YIGM layouts are labeled by the orientation of the sensor first, followed by the component measured. In the case of the tangential orientation of YIGMs, both components showed approximately the same results, so only one of them is presented in the legend. The metrics for Elekta and OPMs were calculated using constant noise levels, so they represent constants as well. The dotted magenta and green vertical lines correspond to the noise level of Elekta (3 $fT/\sqrt{Hz}$) and OPMs (10 $fT/\sqrt{Hz}$), respectively, and are added for convenience.

Among the YIGM layouts, the normally oriented sensors measuring the normal component show the best results, while tangentially oriented YIGMs reveal lower metric values. The lowest metrics can be attributed to normally oriented sensors measuring the tangential component. YIGM layouts show better results than SQUIDs in the case of the 3 $fT/\sqrt{Hz}$ noise level and OPMs in the case of the 10 $fT/\sqrt{Hz}$ noise level in both metrics. However, to outperform the conventional SQUID-based MEG system Elekta Neuromag consisting of 102 magnetometers, we need the intrinsic noise level of YIGMs to be <5 $fT/\sqrt{Hz}$, and <13 $fT/\sqrt{Hz}$ to outperform the OP-MEG developed within the same montage.

The right column in Figure 6 shows the same for gradiometers, taking into account the fact that only one type of YIGG was used: gradiometers registering the tangential gradient of the normal component and oriented normally (presented by the solid blue line). The relative SNR and channel capacity of the YIGGs are compared only with those of the original Elekta’s gradiometer layout (204 planar gradiometers), shown with the bold magenta line since we do not consider gradiometers based on OPMs in this paper. The gradiometer scheme with an assumed noise level of $\sigma_{YIGM}$ = 10 $fT/\sqrt{Hz}$ outperforms SQUID gradiometers in terms of relative SNR and channel capacity. In the case of $\sigma_{YIGM}$ = 3 $fT/\sqrt{Hz}$, the YIGG SQUID-like layout dramatically outperforms SQUIDs gradiometers in terms of both metrics. To achieve the same SNR, SQUID gradiometers need the noise level of YIGG to be <17.5 $fT/\sqrt{Hz}$. At 15 $fT/\sqrt{Hz}$, it starts to outperform Elekta’s gradiometer layout in terms of the total information capacity.

### 5.2. Uniform Layouts

In this section, we compare the uniform layouts formed with YIGMs and described in Section 4.3 with the conventional SQUID-MEG system (Elekta Neuromag) as well as between each other. Since these layouts have different sensor numbers, we can explore how the required noise level changes depending on the number of sensors. As before, in order to compute the relative signal powers and relative SNRs, we use the Elekta Neuromag as the reference system.

The dependencies of metrics on the noise levels for different uniform layouts are shown in Figure 7. The structure of this figure is similar to that of Figure 6. With an increase in the number of sensors in a layout, the metric values grow for both types of magnetometers and gradiometers.

Considering the 333m-nl-nl layout, we start to outperform Elekta Neuromag at the levels of 7.5 $fT/\sqrt{Hz}$ and 9.5 $fT/\sqrt{Hz}$ in terms of RSNR and information capacity, respectively. Similar to SQUID-like layouts, the uniform layouts with tangentially oriented YIGMs show lower metrics compared to normally oriented ones. Moreover, it is interesting to note that for tangential layouts, the difference in RSNR is subtle, and all four curves coincide.

The 333grad-nl-nl layout allows outperforming Elekta’s gradiometers at the noise level of about 30–35 $fT/\sqrt{Hz}$ in terms of the relative SNR and total information capacity.

Figure 8 shows a visual representation of the required number of channels in two types of magnetometric layouts, as well as gradiometer layouts, for a certain intrinsic noise level of YIGM sensors. The red line indicates the same value as for Elekta Neuromag in terms of relative SNR (top row) and information capacity (bottom row). In the case of RSNR, we have zero on the red line since we take the logarithm of 1. The dependencies for relative SNR represent almost straight, perpendicular vertical lines, indicating no explicit dependency between intrinsic noise and the number of channels in the layout.

For additional motivation, to investigate the power of the coverage, we show the relative YIGM/OPM and relative YIGM/Elekta SNR cortical maps for magnetometers in Figure 9. We used three layouts: minimal uniform layout (35 sensors), Elekta-like layout (102 magnetometers), and maximum uniform layout (333 sensors). Since normally oriented YIGMs measuring the normal component performed better, we only used them for these calculations. The noise level of YIGMs was chosen to be equal to that of the comparing type of magnetometer, which was 10 $fT/\sqrt{Hz}$ for OPMs and 3 $fT/\sqrt{Hz}$ for SQUIDs, respectively. The scale ranges from 0.0 to 10.0, where blue color indicates poor coverage, red indicates good coverage, and green represents equal coverage as that of the comparing type of magnetometer. The mean value of the relative SNR is indicated for each brain map. An increase in the number of channels results in better coverage, and the YIGM vs. OPM coverage shows a considerably different result.

Figure 10 shows the same for the gradiometers, except for the scale. Since gradiometers are fundamentally different, the scale here is from 0.5 to 60.

## 6. Discussion

The results presented above may give some hints for further development of the multichannel YIG-MEG system. Here, we present a qualitative analysis of the quantitative results.

Analyzing Figure 6, we conclude that the current noise level of YIGM sensors (35 $fT/\sqrt{Hz}$) is unfortunately insufficient for the construction of a multichannel YIG-MEG system. Although these sensors are closer to the head surface and outperform SQUIDs and OPMs at their noise levels, the overall SNR of a MEG system built with these magnetometers is rather low. The weak dependence of the overall SNR of the sensor array on the number of channels described in Figure 8 is expected, given the assumption of independence of the sensors in an array with respect to each other.

We can conclude that for the densest layout consisting of 333 magnetometers, the YIGM noise level needs to be reduced to one-third of the current level to outperform the Elekta Neuromag in terms of total information capacity. However, as stated before, the SNR is not significantly affected by the number of sensors, and the noise level required to outperform Elekta Neuromag in terms of SNR is still far from the current level.

The computed metrics appear to be better for sensor layouts composed of gradiometers. This is expected, as gradiometers are known to perform better than magnetometers for cortical sources. Cortical maps in Figure 9 and Figure 10 confirm this, as gradiometers provide better coverage. However, it is still evident that the current intrinsic noise level of YIGM sensors is higher than that of SQUIDs.

It is worth noting that, in the case of the most dense uniform YIGG layout (333 gradiometers), the YIGG noise level required to outperform Elekta in terms of relative SNR and total information capacity is very close to the current YIGM noise level.

Speaking of YIGM tangential layouts, as mentioned earlier, we obtained considerably poorer results compared to normal layouts. At first thought, it seems illogical since, in the case of tangential placement, all points on the plane of the YIGM sensor are more or less equidistant from the scalp surface. However, as stated in Section 1.2, the weakness of tangential layouts can be explained by the fact that the magnetic field caused by volumetric currents substantially suppresses the overall amplitude of the topography created by primary currents in tangential measurements [2]. In this way, the decrease in SNR becomes plausible. Further research is required to understand this phenomenon better.

A short comment on YIGM vs. OPM cortical maps: since the comparison between YIGM and OPM was a secondary goal of this paper, we did not devote much attention to it. However, it is clear from Figure 9 that, with the same noise level, YIGM does not provide a considerable advantage in coverage, even with the most dense layout. In contrast, YIGM vs. SQUID maps show that 333 YIGM sensors provide excellent coverage, particularly in the frontal and temporal regions. Additional research is required to determine the cause of this difference.

Despite the above, it is essential to note that, in case of approaching the theoretical limit for YIGM sensors (1 $fT\sqrt{Hz}$), the YIG-MEG system with 102 gradiometers or magnetometers may outperform Elekta Neuromag by 20–60 times in terms of SNR and by 15–40% in terms of channel capacity. Since the approach to the theoretical sensitivity for YIGMs is a combination of engineering and scientific challenges, we assume these values to be a good motivation for further YIGM/YIGG development.

## 7. Conclusions

In this simulation study, we explored different possible layouts for multi-channel YIG-MEG systems and compared the performance of YIGMs with conventional SQUID systems as well as well-established OPMs via lead-field analysis. The values of relative SNR and total information capacity revealed that the primary step towards an efficient multi-channel YIG-MEG system is to reduce the YIGM sensor noise. This is because an increase in the number of channels (decrease in size) with the current intrinsic noise of 35 $fT/\sqrt{Hz}$ does not provide a visible improvement in the YIGM performance.

Our future work on forward modeling will be devoted to calculating optimal sensor locations on the head, using more complex principles than just uniform distribution, and incorporating more realistic source variances. We will also explore the possible use of an averaged head model instead of an individual one. A common practice for EEG and MEG forward problem modeling is the boundary-element method (BEM) (MNE, Brainstorm). Therefore, we plan to try different methods, including more advanced schemes, such as the finite-element method (FEM) [53]. Finally, we will pay special attention to tangential YIGM layouts and their effectiveness.

## Figures and Tables

**Figure 1 sensors-23-04256-f001:**
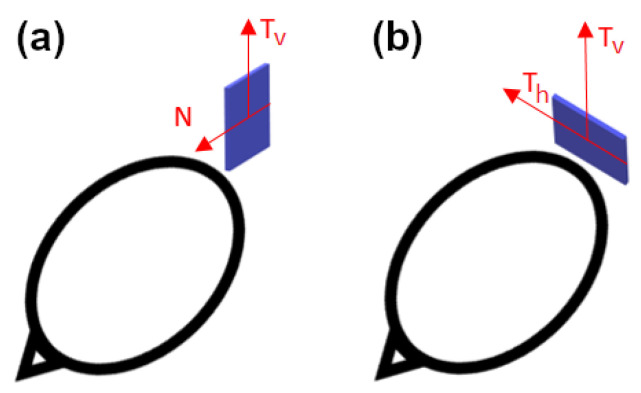
Placement of the YIGM sensor for measurements of radial (**a**) and tangential (**b**) components of the magnetic induction.

**Figure 2 sensors-23-04256-f002:**
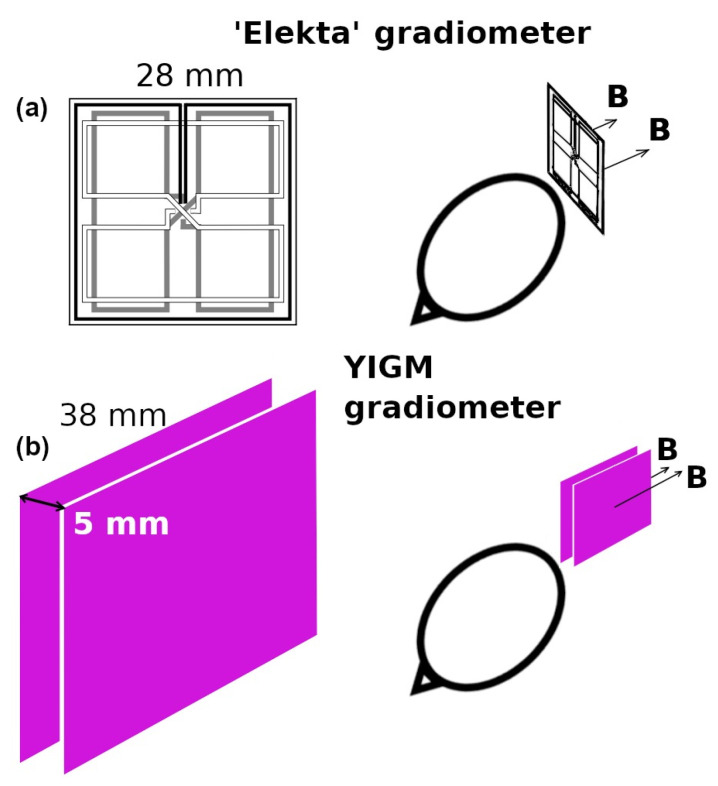
(**a**) Elekta’s gradiometer scheme and (**b**) representation of the YIGM gradiometer scheme for measuring tangential derivatives of the radial component of the magnetic induction vector ($\frac{\partial B_{n}}{\partial x}$).

**Figure 3 sensors-23-04256-f003:**
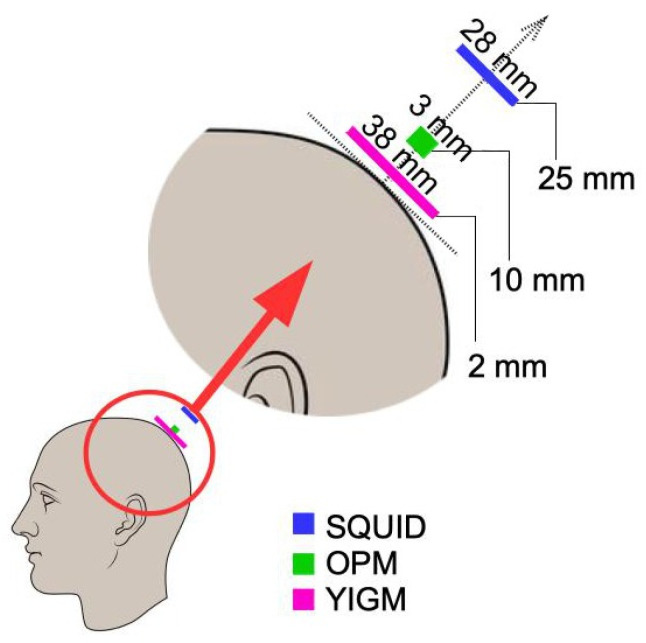
Positions of sensors of different types with respect to the head surface.

**Figure 4 sensors-23-04256-f004:**
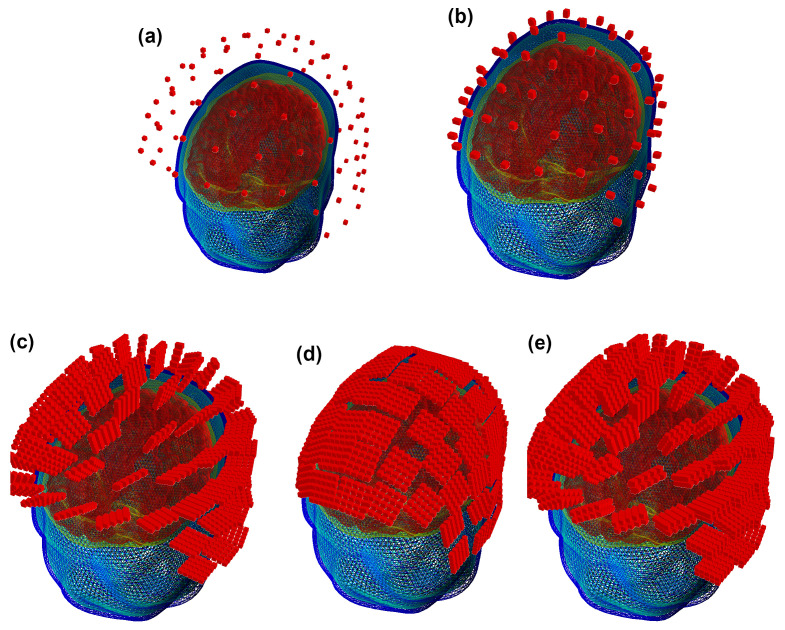
SQUID-like layouts: (**a**) original SQUID layout (Elekta Neuromag MEG system); (**b**) OPM SQUID-based layout; (**c**) YIGM-NL (normally oriented) SQUID-based layout; (**d**) YIGM-TG (tangentially oriented) SQUID-based layout; (**e**) YIGG-NL (gradiometer scheme with sensors normally oriented) SQUID-based layout.

**Figure 5 sensors-23-04256-f005:**
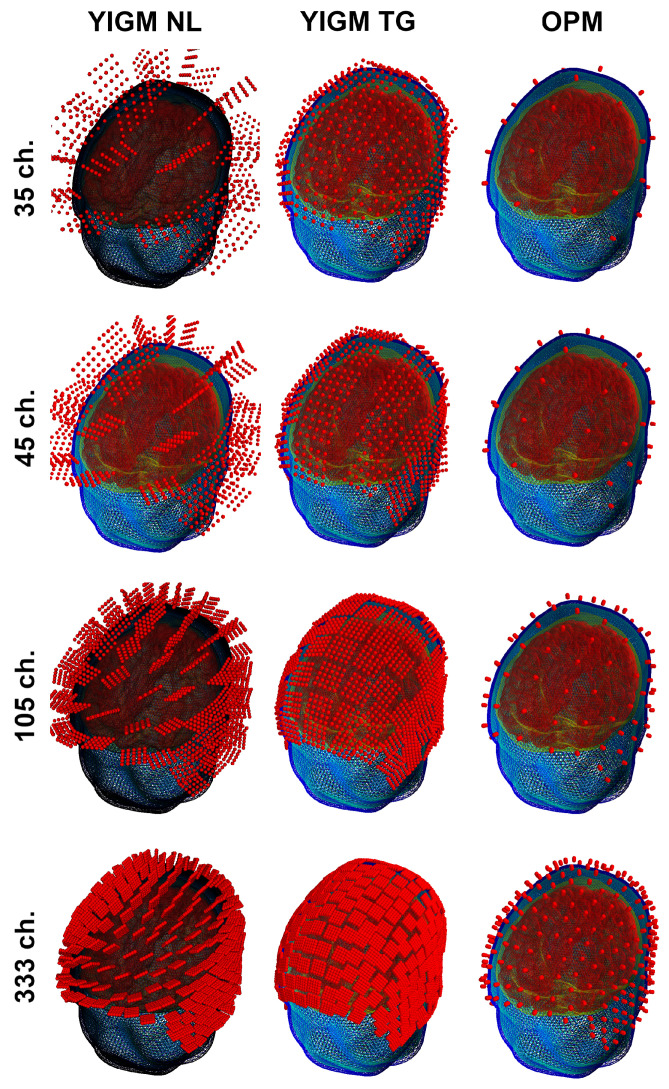
Uniform layouts corresponding to Table 2. The number of sensors for each layout triplet (row) is shown on the left side.

**Figure 6 sensors-23-04256-f006:**
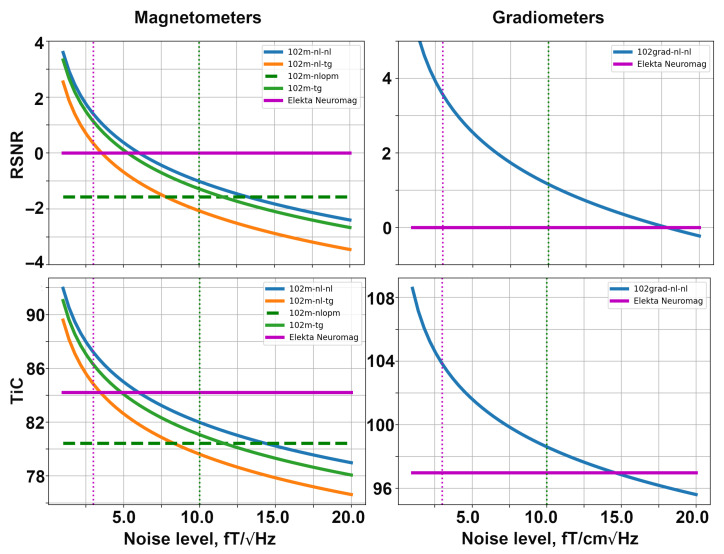
Dependencies of relative SNR (top row) and total information capacity (bottom row) for SQUID-based layouts formed with magnetometers (**left**) and gradiometers (**right**).

**Figure 7 sensors-23-04256-f007:**
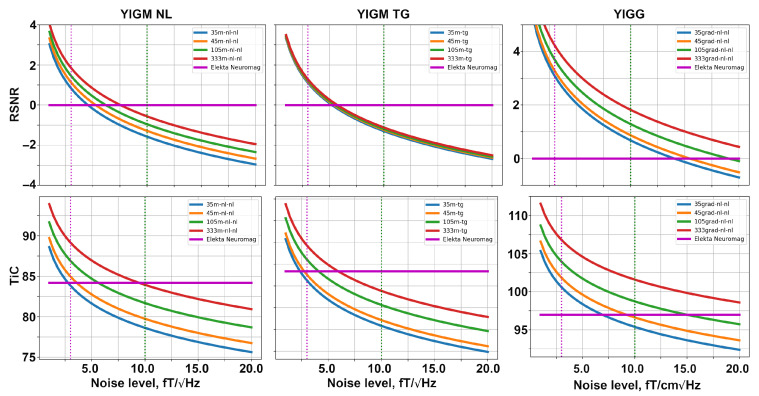
Relative SNR (top row) and total information capacity (bottom row) computed for different uniform layouts of YIGM: magnetometers (**left and middle columns**), and gradiometers (**right column**).

**Figure 8 sensors-23-04256-f008:**
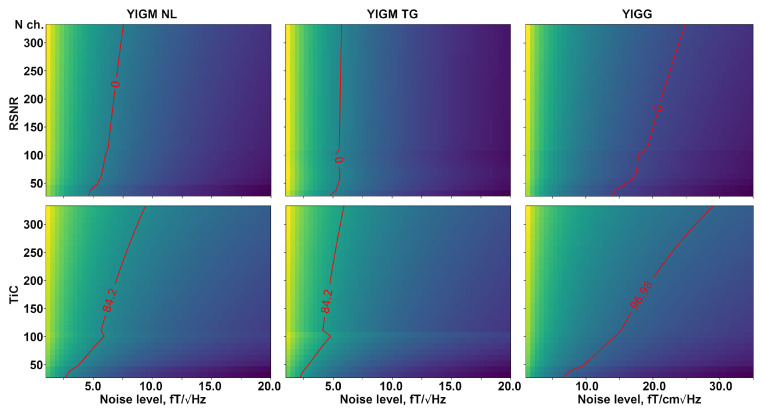
Dependence of relative SNR (top row), and total information capacity (bottom row) on the number of channels and the different intrinsic noise levels compared to the conventional Elekta Neuromag SQUID-MEG system. The left column represents normally oriented YIGMs, the middle column represents tangentially oriented YIGMs, and the right column shows the metrics for normally oriented YIGM-based gradiometers with a base of 5 mm. The red line represents the equal metric value to that of the reference SQUID-MEG Elekta Neuromag system.

**Figure 9 sensors-23-04256-f009:**
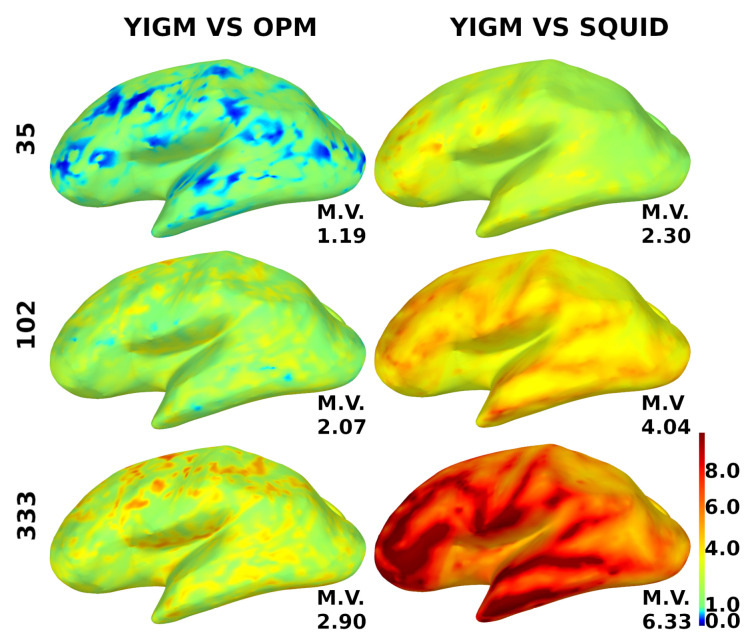
Distribution of the relative SNR over the cortex for magnetometers. YIGM layouts (minimum uniform, SQUID-like, and maximum uniform) with noise level $\sigma_{YIGM}$ = $\sigma_{OPM}$ = 10 $fT/\sqrt{Hz}$ compared with OPMs (left panel) and with noise level $\sigma_{YIGM}$ = $\sigma_{SQUID}$ = 3 $fT/\sqrt{Hz}$ compared with SQUID magnetometers (right panel). The scale is from 0.0 to 10.0.

**Figure 10 sensors-23-04256-f010:**
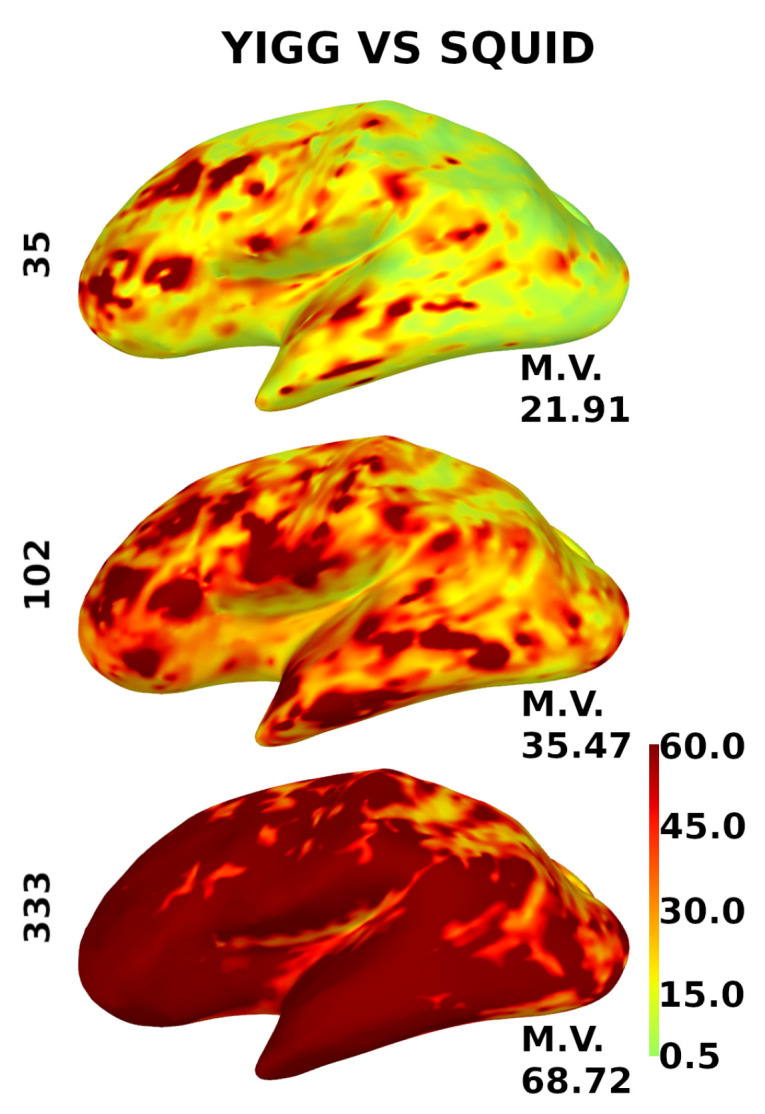
Distribution of the relative SNR over the cortex for gradiometers. YIGM layouts (minimum uniform, SQUID-like, and maximum uniform) all with noise levels $\sigma_{YIGM}$ = $\sigma_{SQUID}$ = 3 $fT/cm\sqrt{Hz}$ compared with SQUID gradiometers. The scale is from 0.5 to 60.

**Table 1 sensors-23-04256-t001:** Parameters of all sensors used in this study (from left to right): intrinsic noise σ, distance to scalp *d*, number of integration points $N_{i}$, base, and size.

Sensor Type	$\sigma ,\mathrm{fT}/\sqrt{\mathrm{Hz}}$	*d*, mm	$N_{i}$	Base, mm	Size
OPM	10 [23]	10.0	24	N/A	$3^{3}$ mm^3^
SQUID magnetometer	3	25.0	16	N/A	${\left(2.1\ldots 2.8\right)}^{2}$ mm^2^
YIGM magnetometer (rad.)	1...35	2.0	25	N/A	${\left(10\ldots 38\right)}^{2}$ mm^2^
YIGM magnetometer (tang.)	1…35	2.0	25	N/A	${\left(10\ldots 38\right)}^{2}$ mm^2^
**Sensor Type**	$\sigma ,\mathrm{fT}/\mathrm{cm}\sqrt{\mathrm{Hz}}$	d, **mm**	$N_{i}$	**Base, mm**	**Size**
SQUID gradiometer	3	25.0	8	16.8	$2.639^{2}$ mm^2^
YIGM gradiometer	1…35	2.0	50	5.0	${\left(10\ldots 38\right)}^{2}\times 5$ mm^3^

**Table 2 sensors-23-04256-t002:** Uniform layouts: number of sensors $N_{s}$ depending on the sensor sizes. The OPM sizes are not shown since they are constant.

Size, mm^2^	38^2^	30^2^	20^2^	10^2^
$N_{s}$	35	45	105	333

## Data Availability

The data that support the findings of this study are available from the corresponding author upon reasonable request.

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
