# Peer review of "Yttrium-Iron Garnet Magnetometer in MEG: Advance towards Multi-Channel Arrays"

_sensors, 2023, doi:10.3390/s23094256_

Round 1

Reviewer 1 Report

In the manuscript, the Authors evaluate the performance of various MEG sensor arrays based on yttrium-iron garnet magnetometers (YIGMs). The performances of these arrays are compared to state-of-the-art SQUID and OPM arrays. The Authors make the right conclusion that the current noise level of the YIGM is limiting the performance of these arrays compared to the SQUID and OPM arrays. However, by improving the noise level, the YIGM arrays should provide good performance due to their closeness to the subject’s scalp.

I enjoyed reading the manuscript and did not find any significant flaws in the methodology. However, several major and minor issues have to be addressed in the manuscript. Below I list my comments about the manuscript in detail.

Major comments

Abstract: “The results showed that YIGMs outperform SQUIDs and OPMs at their noise level.” This sentence is too vague. It is not clear to what “their” refers to. Please rephrase. Please also describe why they outperform OPMs and SQUIDs.

In introductory Section 1.1., please give sensitivity levels for different sensors (high Tc and low Tc SQUIDs, OPMs, YIGM etc). For example, I think that the following sentence related to high Tc  SQUIDs: “They operate at liquid nitrogen temperature instead of helium, having sensitivity close to that of conventional low-Tc SQUIDs” is not correct. High Tc SQUIDs have much higher noise level than low Tc SQUIDs. Please give values for the sensitivity levels in the text.

Sec. 1.2.: “Iivanainen et al. [2] shows, e.g. the greater impact of volumetric Ohmic currents on the tangential component of the magnetic induction compared to that on radial component. The works [28] and [29] show, however, absence of theoretical grounds of this phenomenon,”. Please elaborate here. How the works [28] and [29] indicate the absence of the theoretical ground of this phenomenon? Please explain.

Sec. 3.2.: “Adapted for MEG/EEG measurements by J.Schneiderman, the total information capacity is related to original Shannon’s channel capacity (see [31])”. The total information capacity or channel capacity was adapted for MEG measurements first time by Kemppainen and Ilmoniemi (1989).

Kemppainen, P. K., & Ilmoniemi, R. J. (1989). Channel capacity of multichannel magnetometers. Advances in biomagnetism, 635-638.

Sec. 3.2.: “Information capacity”. How was the source variance q^2 set in the simulations?

Table 1: Several comments:

·       Please give references to the OPM sensor sensitivity level of 10 fT/rHz.

·       How were the 24 integration points defined for the OPM?

·       Unit of the sensitivity of SQUID gradiometer should be 3/fT/rHz/cm. What were the gradiometer units used in the simulation?

·       YIGM gradiometer size: “(10..38)^2 × 5 mm^2”. Is there a mistake in “× 5”?

Results section: How is the relative signal power calculated/defined? I do not think that it should depend on the sensor noise level.

Figure 6: What was the YIGM size in the simulation shown in Fig. 6? Why are the RP, RSNR, TiC values for 102m-nl-nl higher than for OPM at 10 fT/rHz? Please provide an explanation.

Sec. 5.1. ”The gradiometric scheme with assumed noise level σYIGM = 10 f T/√Hz outperforms both SQUID gradiometers and OPMs in terms of relative signal power and channel capacity, but shows lower SNR.” I do not think that you are comparing gradiometric YIGM to OPM in the simulations you show.

Mistake: Figures 9 and 10 are the same.

Minor comments

Abstract: “signal-to-noise ration (SNR)”. ratio

Sec 1.1.: “The main technological difficulty of the MEG caused by” is caused by

Sec. 1.1.: “Yttrium-iron garnet magnetometer (YIGM) is a solid-state sensor and unlike SQUIDs and OPMs operates at room temperatures”. Some OPMs operate at room temperature, for example, OPMs based on He4 isotope.

Equation 3: What is Delta B(x)?

Sec. 2.2.: “The number of source defines its position,” I did not understand this.

Sec. 2.2.: “The inverse problem is ill-posed according to Hadamard due to its solution being unstable” Please clarify.

Sec. 2.2.: “Due to [17,43], the regularization parameter may be computed using the estimated signal-to-noise ratio (SNR)” Due to what and why in [17,43]? Please elaborate verbally.

Sec. 3.: “. In this case, the noise covariance matrix CYIGM takes the form of diagonal matrix with dispersion of intrinsic noise σYIGM at the diagonal.” The diagonal of a covariance matrix consists of variances, so in the sentence σYIGM should be squared: σYIGM^2.

Sec. 3.1.: Indeed, by introducing the value q of source variance for independent sources”. Variances are squared so q should be q^2.

Include references to Eqs. 9, 10 and 11.

Sec. 5: “The source power for each layout was computed using the column-wise norm of the lead-field matrix” column-wise squared norm?

Figure 6 y-label: “Magnitometers” typo

Figure 6: Please describe how the values are normalized in the plots. It seems that they are normalized with respect to SQUIDs.

Fig. 9 caption:  “compared with SQUID gradiometers (right panel).” gradiometers -> magnetometers.

Figure 5 caption: “Tab. 2” -> Table 2

Sec. 7.: “Moreover, in this paper we used MNE, which in its turn involves the boundary-element method (BEM) for MEG modeling.” Contrary to what is said in the previous sentence, MNE does not always involve BEM.

Reviewer 2 Report

Dear Authors,
You present an interesting simulation study of a new promising solid-state yttrium-iron garnet magnetometer (YIGM) for the detection of magnetoencephalographic (MEG) signals and compare them with well-established MEG systems based on superconducting quantum interference devices (SQUIDs) and optically-pumped magnetometers (OPMs). The study is well-designed, but some parts of the methods and results are not well described and have to be clarified and improved.

Main objections:
1) The description of the source space you used in the lead-field matrix calculation is missing. You briefly mentioned the construction of the lead-field matrix in section 2 (Theory, lines 167-9) and how you segmented the cortex matter in section 3 (Methods, lines 210-4). However, there is no information about the source space, i.e., the total number of dipoles per hemisphere, their positions and orientations.

2) It is unclear how the relative signal power (RP) depends on the noise you showed as the results in the left columns of Figs. 6, 7, and 8. According to the definition in Eq. (11) and the description in lines 227-8, the signal power is calculated as the l2-norm of the source topography squared, where the source topography is defined by columns of the lead-field matrix (lines 216-8). But the lead-field matrix depends only on the geometry of source space and sensor system, i.e., positions of sources and sensors and sensing direction.
Please, explain in more detail how you calculate RP and how it depends on noise, or show only RSNR and TiC as results.

3) Wrong images in Fig. 10, which are the same as in Fig. 9. I guess it is a reformatting error because the preprint of this article was published in a slightly different format in bioRxiv in July 2022, https://doi.org/10.1101/2022.07.11.499607, where images in  Fig. 10 are correct.

Minor objection:
4) I found typing error in line 63 (banch --> bunch), the sentence should be:
Along with the rise of new types of sensors a bunch of simulation works appear to find and explore new layouts [16-18].
You might also consider citing here some other recent simulation works:
Beltrachini, et al. Optimal Design of On-scalp Electromagnetic Sensor Arrays for Brain Source Localisation. Hum. Brain Mapp. 2021, 42, 4869–4879, https://doi.org/10.1002/hbm.25586
U. Marhl, et al. Simulation Study of Different OPM-MEG Measurement Components. Sensors 2022,  22, 3184, https://doi.org/10.3390/s22093184

Reviewer 3 Report

This paper introduces a recently-developed MEG-sensor, yttrium-iron garnet magnetometer (YIGM). The authors calculates the lead-field matrices of several types of YIGM sensor arrays and created their SNR maps, which were compared with the conventional SQUID-based MEG and optically-pumped magnetometers (OPMs).The results showed that YIGMs outperform SQUIDs and OPMs at their noise level.

This is an interesting simulation and may lead to a further development of this new technique. My only suggestion is that it would be of interest to show the absolute SNR in addition to the relative SNR maps presented in the manuscript. The readers would be already familiar with these maps of SQUID and OPMs, so only YIGM maps would be of interest.

Round 2

Reviewer 1 Report

I want to thank the Authors for revising the manuscript. I still found one small mistake in the manuscript. I think that the units in the caption of figure 10 should be fT/rHz/cm. Otherwise I think that the manuscript is ready for publication.